# Density of States Prediction of Crystalline Materials via Prompt-guided Multi-Modal Transformer

**Namkyeong Lee**[1†], **Heewoong Noh**[1†], **Sungwon Kim**[1],
**Dongmin Hyun**[2], **Gyoung S. Na**[3], **Chanyoung Park**[1*]
[1] KAIST      [2] Yahoo Research      [3] KRICT
{namkyeong96,heewoongnoh,swkim,cy.park}@kaist.ac.kr
dhyun@yahooinc.com, ngs0@krict.re.kr

## Abstract

The density of states (DOS) is a spectral property of crystalline materials, which provides fundamental insights into various characteristics of the materials. While previous works mainly focus on obtaining high-quality representations of crystalline materials for DOS prediction, we focus on predicting the DOS from the obtained representations by reflecting the nature of DOS: *DOS determines the general distribution of states as a function of energy.* That is, DOS is not solely determined by the crystalline material but also by the energy levels, which has been neglected in previous works. In this paper, we propose to integrate heterogeneous information obtained from the crystalline materials and the energies via a multi-modal transformer, thereby modeling the complex relationships between the atoms in the crystalline materials and various energy levels for DOS prediction. Moreover, we propose to utilize prompts to guide the model to learn the crystal structural system-specific interactions between crystalline materials and energies. Extensive experiments on two types of DOS, i.e., Phonon DOS and Electron DOS, with various real-world scenarios demonstrate the superiority of DOSTransformer. The source code for DOSTransformer is available at https://github.com/HeewoongNoh/DOSTransformer.

## 1   Introduction

Along with recent advances in machine learning (ML) for scientific discovery, ML models have rapidly been adopted in computational materials science thanks to the abundant experimental and computational data in the field. Most ML models developed in computational materials science to date have been focused on crystalline materials property consisting of single-valued quantities [29], e.g., band gap energy [31, 69], formation energy [3, 56], and Fermi energy [61].

On the other hand, spectral properties are ubiquitous in materials science, characterizing various properties of crystalline materials, e.g., X-ray absorption, dielectric function, and electronic density of states [29] (Figure 1).

The density of states (DOS), which is the main focus of this paper, is a spectral property that provides fundamental insights on various characteristics of crystalline materials, even enabling direct computation of single-valued properties [17]. For example, DOS is utilized as a feature of materials for analyzing the underlying reasons for changes

Figure 1: A crystal structure and various types of its properties.

---

*Corresponding author.   † These authors contributed equally.

37th Conference on Neural Information Processing Systems (NeurIPS 2023).

in electrical conductivity [15]. Moreover, band gaps and edge positions, which can be directly derived from DOS, are utilized to discover new photoanodes for solar fuel generation [47, 64]. As a result, the exploration of ML capabilities for predicting DOS takes us one step further toward the core principles of materials science, thereby expediting the process of materials discovery. However, credible computation of DOS with traditional density functional theory (DFT) requires expensive time/financial costs of exhaustively conducting experiments with expertise knowledge [7, 14]. Therefore, alternative approaches for DOS calculation are necessary, whereas demonstrating ML capabilities for learning such spectral properties of the crystalline materials is relatively under-explored.

In this paper, we focus on predicting DOS by following the nature of DOS calculation: *DOS determines the general distribution of states as a function of energy*. That is, while single-valued properties can be described solely by the crystalline material, DOS is determined by not only the material itself but also the *energy levels* at which DOS is calculated. Therefore, integrating the heterogeneous signals from both the crystalline material and the various energy levels is crucial for DOS prediction. However, previous works for DOS prediction focus on obtaining a qualified representation of material [7, 14, 17, 9], thereby overlooking the unique characteristics of DOS.

On the other hand, learning from multiple input types, i.e., crystalline material and various energy levels, is not trivial due to their heterogeneity. To this end, we formulate the DOS prediction problem into a multi-modal learning problem, which has recently gained significant attention from ML researchers in various domains thanks to its capability of extracting and relating information from heterogeneous data types [6, 38, 54, 5, 2]. For example, VisualBERT proposes to align elements of input modalities, i.e., images and text, thereby achieving SOTA performance in various vision-and-language tasks [36]. Moreover, Dall-E shows its power in generating images based on text data by encoding images and texts together [43]. Among various multi-modal learners, the multi-modal transformer demonstrates its intrinsic advantages and scalability in modeling multiple modalities [62], by associating heterogeneous modalities with the cross-attention mechanism [65, 50].

Inspired by the recent success of the multi-modal transformer, we propose a multi-modal transformer model for DOS prediction, named DOSTransformer, which incorporates crystalline material and energies as heterogeneous input modalities. Unlike previous works that overlook the energy levels for predicting DOS [7, 14, 17, 9], DOSTransformer learns embeddings of energy levels which are used for modeling complex relationships between the energy levels and crystalline material through a cross-attention mechanism. By doing so, DOSTransformer obtains multiple representations for a single crystalline material according to various energy levels, enabling the prediction of a single DOS value on each energy level. However, simply combining a crystalline material with energy levels may fail to consider the significant impact of the crystal's structure, which sometimes results in distinct material properties even when it is composed of identical elements (e.g., carbon). For example, although graphite and diamond are made entirely out of carbon, their arrangement (or structure) of carbon atoms differs significantly, resulting in distinct material properties. Therefore, we utilize learnable prompts to inform DOSTransformer of the input crystal system (among the seven crystal systems[2]), which navigates the model to capture structure-specific interactions between a crystalline material and energy levels. By doing so, DOSTransformer learns to extract not only the relational information that is shared with a crystal system, but also across the crystal systems. In this work, we make the following contributions:

- Inspired by the fact that DOS is determined by not only the materials themselves but also the energy levels at which DOS is calculated, we propose a multi-modal transformer model for DOS prediction, called DOSTransformer, which incorporates crystalline materials and energy as heterogeneous input modalities.

- To capture structure-specific interactions between a crystalline material and energy levels, thereby being able to predict DOS that significantly differs according to the crystal's structure, we utilize learnable prompts to inform DOSTransformer of the input crystal system.

- Our extensive experiments on two types of DOS, i.e., Phonon DOS and Electron DOS, demonstrate that DOSTransformer outperforms a wide range of state-of-the-art methods in various real-world scenarios, i.e., in-distribution scenarios and out-of-distribution scenarios.

To the best of our knowledge, this is the first work that considers various energy levels during DOS prediction and introduces prompts for crystal structural systems.

---

[2]https://en.wikipedia.org/wiki/Crystal_system

## 2 Related Works

### 2.1 Machine Learning for Materials and Density of States

Traditional materials science research heavily relies on theory, experimentation, and computer simulations, but these approaches are time-consuming, costly, and inefficient, hindering their ability to keep up with the field's development [57]. As an alternative research tool, ML, which requires neither expensive trial and error experiments nor domain expert knowledge, has been attracting a surge of interest from materials scientists [67]. Consequently, various ML methods have been established to learn the relationship between materials and their properties which are usually obtained from ab-initio calculations [11]. Inspired by the recent success of GNNs in biochemistry [19, 48, 25, 33, 34, 32], CGCNN proposes a message-passing framework for crystal structure providing high-accuracy prediction for 8 different materials properties [61]. However, most studies have focused on single-valued properties [29], whereas various spectral properties, such as DOS, are also crucial.

The DOS describes the number of different states at a particular energy level that particles are allowed to occupy, determining various properties of crystalline materials. For example, phonon DOS has a crucial influence in determining the specific heat and vibrational entropy of crystalline materials as well as the interfacial thermal resistance [60], while electron DOS is used to derive the electronic contribution to heat capacity in metals [21] and the effective mass of electrons in charge carriers [18].

Despite the importance of DOS, how to incorporate ML capabilities for *decoding* such spectral properties is under-explored since the vast majority of methods concentrate only on *encoding* crystalline materials. Specifically, Chandrasekaran et al. [7] proposes to predict electron DOS based on the hand-crafted fingerprint of each grid point in the crystalline materials, while Del Rio et al. [14] leverages that of each atom in the crystalline materials. Chen et al. [9] predicts phonon DOS by learning the representations of the crystalline materials that are equivariant to 3D rotations, translations, and inversion with Euclidean neural networks [49, 28, 58], achieving high-quality prediction with a small number of training materials with over 64 atom types. In this work, we concentrate on decoding spectral properties regarding the complex relationship between the various energy levels and crystalline materials.

### 2.2 Multi-modal Transformer, Prompt Tuning, and Positional Encoding

**Multi-modal Transformer.** In recent years, much progress in multi-modal learning has grown rapidly by extracting and aligning the rich information from heterogeneous modalities [2, 6]. Among the various multi-modal learning methods, the multi-modal transformer demonstrates its superior capability in learning multiple modalities [62], by associating the heterogeneous modalities with a cross-attention mechanism. Specifically, MulT [50] learns the representations directly from unaligned multi-modal streams with multi-modal transformer, while Yao & Wan [65] improves machine translation quality by learning the image-aware text representations. In this paper, we investigate how to decode DOS of a crystalline material by learning the energy-aware crystal representations via a multi-modal transformer.

**Prompt Tuning.** On the other hand, prompt designing has gained significant attention from NLP researchers as an efficient method for fine-tuning large language models (LLMs) for various tasks [39]. There are two main approaches to prompt designing: discrete prompt designing [44] and continuous prompt tuning [37]. While discrete prompt designing involves manually creating task description tokens for LLMs through trial and error, continuous prompt tuning trains learnable prompts in a continuous latent space without requiring expert knowledge. Inspired by the success of continuous prompt tuning in NLP domain, it has been widely adopted in various domains such as computer vision [24] and vision-language [68]. While previous works utilize prompts for efficient fine-tuning of the models, we employ prompts to capture the heterogeneity of various crystal structures.

**Positional Encoding.** In the original proposal of the Transformer architecture, a sinusoidal function was suggested as a method for positional encoding. However, the sinusoidal function may have limitations in terms of learnability and flexibility, which can affect its effectiveness [53]. To address this issue, most pre-trained language models [16, 40] employ learnable vector embeddings as positional representations. By conceptualizing each energy value as the position of a word in a sentence, various energy embeddings in Section 4.2 can be analogous to the positional encoding used in traditional transformers.

# 3 Preliminaries

**Notations.** Let $\mathcal{G} = (\mathcal{V}, \mathcal{A})$ denote a crystalline material, where $\mathcal{V} = \{v_1, \ldots, v_n\}$ represents the set of atoms, and $\mathcal{A} \subseteq \mathcal{V} \times \mathcal{V}$ represents the set of edges connecting the atoms in the crystalline material $\mathcal{G}$. Moreover, $\mathcal{G}$ is associated with a feature matrix $\mathbf{X} \in \mathbb{R}^{n \times F}$ and an adjacency matrix $\mathbf{A} \in \mathbb{R}^{n \times n}$ where $\mathbf{A}_{ij} = 1$ if and only if $(v_i, v_j) \in \mathcal{A}$ and $\mathbf{A}_{ij} = 0$ otherwise.

**Task: Density of States Prediction.** Given a set of crystalline materials $\mathcal{D}_{\mathcal{G}} = \{\mathcal{G}_1, \mathcal{G}_2, \ldots, \mathcal{G}_N\}$ and a set of energies $\mathcal{D}_{\mathcal{E}} = \{\mathcal{E}_1, \mathcal{E}_2, \ldots, \mathcal{E}_M\}$, our goal is to train a model $\mathcal{M}$ that predicts the DOS of a crystalline material given a set of energies, i.e., $\mathbf{Y}^i = \mathcal{M}(\mathcal{G}_i, \mathcal{D}_{\mathcal{E}})$, where $\mathbf{Y}^i \in \mathbb{R}^M$ is an $M$ dimensional vector containing the DOS values of a crystalline material $\mathcal{G}_i$ at each energy $\mathcal{E}_1, \ldots, \mathcal{E}_M$, and $\mathbf{Y}_j^i \in \mathbb{R}$ is the DOS value of $\mathcal{G}_i$ at energy level $\mathcal{E}_j$.

# 4 Methodology: DOSTransformer

In this section, we introduce our proposed method, named DOSTransformer, a novel DOS prediction framework that learns the complex relationship between the atoms in the crystalline material and various energy levels by utilizing a cross-attention mechanism of the multi-modal transformer.

## 4.1 Crystalline Material Encoder

Before modeling the pairwise interaction between the crystalline material and the energies, we first encode the crystalline material with GNNs to learn the representation of each atom, which contains not only the feature information but also the structural information. Formally, given a crystalline material $\mathcal{G} = (\mathbf{X}, \mathbf{A})$, we generate an atom embedding matrix for the crystalline material as follows:

$$\mathbf{H} = \text{GNN}(\mathbf{X}, \mathbf{A}), \tag{1}$$

where $\mathbf{H} \in \mathbb{R}^{n \times d}$ is an atom embedding matrix for $\mathcal{G}$, whose $i$-th row indicates the representation of atom $v_i$, and we stack $L'$ layers of GNNs. Among various GNNs, we adopt graph networks [4] as our crystalline material encoder, which is a generalized and extended version of various GNNs. We provide further details on the GNNs in Appendix C.

## 4.2 Prompt-based Multi-modal Transformer

After obtaining the atom embedding matrix $\mathbf{H}$, we aim to capture the relationship between a crystalline material and various energy levels by utilizing the cross-attention layers of a multi-modal transformer [62] with self-attention layers [52]. In a nutshell, we expect the cross-attention layers to integrate the heterogeneous signals from a crystalline material and various energy levels, while self-attention layers aim to integrate the information of material-specific energy representations.

**Cross-Attention Layers.** Specifically, we expect the cross-attention layers to generate the material-specific representation of the energies by repeatedly reinforcing the energy representation with the crystalline materials. To do so, we first introduce a learnable energy embedding matrix $\mathbf{E}^0 \in \mathbb{R}^{M \times d}$, whose $j$-th row, i.e., $\mathbf{E}_j^0$, indicates the embedding of energy $\mathcal{E}_j \in \mathcal{D}_{\mathcal{E}}$. Then, we present cross-modal attention for fusing the information from the crystal structure into various energy levels as follows:

$$\begin{aligned} \mathbf{E}^l &= \text{Cross-Attention}(\mathbf{Q}_{\mathbf{E}^{l-1}}, \mathbf{K}_{\mathbf{H}}, \mathbf{V}_{\mathbf{H}}) \in \mathbb{R}^{M \times d} \\ &= \text{Softmax}(\frac{\mathbf{E}^{l-1}\mathbf{H}^\top}{\sqrt{d}})\mathbf{H}, \end{aligned} \tag{2}$$

where $l = 1, \ldots, L_1$ indicates the index of the cross-attention layers. In contrast to the conventional Transformer, which introduces learnable weight matrices for query $\mathbf{Q}$, key $\mathbf{K}$, and value $\mathbf{V}$, we directly employ the previously obtained energy embedding matrix $\mathbf{E}^{l-1}$ as the query matrix, and the atom embedding matrix $\mathbf{H}$ as the key and value matrices. Based on the above cross-attention mechanism, we obtain the material-specific energy embedding $\mathbf{E}^l \in \mathbb{R}^{M \times d}$ by aggregating the information regarding the atoms in the crystalline material that were important at each energy level. The final material-specific energy embedding matrix $\mathbf{E}^{L_1} \in \mathbb{R}^{M \times d}$ generated by the cross-attention layer reflects the relationship between the atoms in the crystalline material and various energy levels.

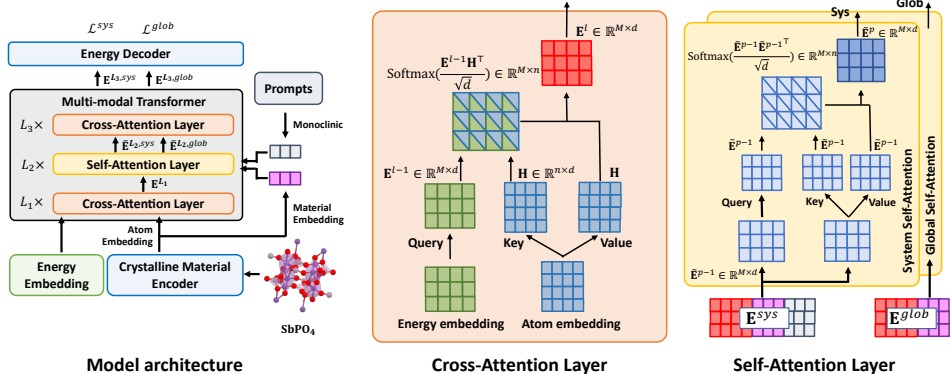

Figure 2: Overall model architecture and attention layers in prompt-based multi-modal Transformer.

**Global Self-Attention.** In addition to cross-attention layers, we propose to enhance the material-specific energy embedding matrix $\mathbf{E}^{L_1}$ by aggregating the information from other energy levels. To do so, we employ variants of self-attention layers in conventional Transformer [52] as follows:

$$
\begin{aligned}
\tilde{\mathbf{E}}^p &= \text{Self-Attention}(\mathbf{Q}_{\tilde{\mathbf{E}}^{p-1}}, \mathbf{K}_{\tilde{\mathbf{E}}^{p-1}}, \mathbf{V}_{\tilde{\mathbf{E}}^{p-1}}) \in \mathbb{R}^{M \times d} \\
&= \text{Softmax}\Big(\frac{\tilde{\mathbf{E}}^{p-1}\tilde{\mathbf{E}}^{p-1\top}}{\sqrt{d}}\Big)\tilde{\mathbf{E}}^{p-1},
\end{aligned}
\tag{3}
$$

where $p = 1, \ldots, L_2$ indicates the index of the self-attention layer. We use an enhanced material-specific energy embedding $\tilde{\mathbf{E}}^0$ as an input of the self-attention, which is obtained by concatenating the material-specific energy embedding $\mathbf{E}^{L_1}$ with the material representation $\mathbf{g}_i \in \mathbb{R}^d$, i.e., $\tilde{\mathbf{E}}^0_j = \phi_1(\mathbf{E}^{glob}_j)$, where $\mathbf{E}^{glob}_j = (\mathbf{E}^{L_1}_j || \mathbf{g}_i)$, $\phi_1 : \mathbb{R}^{2d} \to \mathbb{R}^d$, and $||$ indicates the concatenation operation. Note that $\mathbf{g}_i$ is a sum pooled representation of the material $\mathcal{G}_i$, and $\mathbf{E}^{L_1}_j$ indicates the $j$-th row of the energy embedding matrix $\mathbf{E}^{L_1}$ computed by Equation 2 where $j = 0, \ldots, M$. Although the output of cross-attention layers, i.e., $\mathbf{E}^{L_1} \in \mathbb{R}^{M \times d}$, captures the local atom-level information regarding the material, we also recognize the importance of incorporating the global material-level information $\mathbf{g}_i$ that may not have been fully encoded during the cross-attention phase. Note that the importance of considering the global contextual information for enhancing the learning of dependencies among neural representations has been widely studied in various domains such as computer vision and natural language processing [51, 55, 22]. By additionally considering the global material-level information, we aim to enhance the material-specific representation of the energies more comprehensively.

**System Self-Attention with Crystal System Prompts.** However, solely aggregating the information of material-specific energy embeddings may neglect the crucial influence of the structural properties in crystalline materials, whereas the structural properties play a significant role in determining the unique characteristics and properties of these materials [45]. To illustrate this point, let's examine the case of graphite and diamond, both composed entirely of carbon atoms. In graphite, carbon atoms are arranged in stacked layers, where each carbon atom is bonded to three neighboring carbon atoms in a hexagonal lattice. In contrast, the carbon atom in diamond forms strong covalent bonds with four neighboring carbon atoms, resulting in a tetrahedral arrangement. Even though graphite and diamond are made entirely out of carbon, due to the structural difference, graphite exhibits properties such as electrical conductivity and lubricity due to its layered structure [59, 26], while diamond is renowned for its hardness and thermal conductivity owing to its tightly bonded tetrahedral network [8, 27].

Despite the importance of structural information, effectively incorporating it into the model is not trivial. Naively concatenating the structural information as an input feature of materials is straightforward but may lead to interference, hindering the model's ability to learn and generalize knowledge across different crystal structures. To this end, we propose to provide additional information to the self-attention layers with learnable prompts, which indicate the structural information of the crystalline materials. Specifically, we adopt learnable prompts $\mathbf{P} \in \mathbb{R}^{7 \times d_p}$, whose $k$-th row $\mathbf{P}_k$ represents one of the seven widely known crystal systems: Cubic, Hexagonal, Tetragonal, Trigonal, Orthorhombic, Monoclinic, and Triclinic. Then, given a crystalline material $\mathcal{G}_i$, whose crystal system is given as $k$, we incorporate the learnable prompts into the material-specific energy embedding $\tilde{\mathbf{E}}^0$ as an input

of the self-attention, which is obtained as follows: $\tilde{\mathbf{E}}_j^0 = \phi_2(\mathbf{E}_j^{sys})$, where $\mathbf{E}_j^{sys} = (\mathbf{E}_j^{L_1}||\mathbf{g}_i||\mathbf{P}_k)$ and $\phi_2 : \mathbb{R}^{2d+d_p} \to \mathbb{R}^d$. By doing so, we obtain high-level energy features that contain the crystal structural system information (i.e., $\tilde{\mathbf{E}}^{L_2}$), while keeping the low-level features to share the knowledge across all the materials (i.e., $\mathbf{E}^{L_1}$). In Section 5.4, we demonstrate the effectiveness of our prompt-based approach compared to other variants.

Besides the self-attention layers, we use additional cross-attention layers on top of the self-attention layers, whose input energy embedding $\mathbf{E}^0$ in Equation 2 is given as $\tilde{\mathbf{E}}^{L_2}$, and output the final material-specific energy embeddings $\mathbf{E}^{L_3}$, where $L_3$ is the number of additional cross-attention layers. By doing so, the model extracts the enhanced relationships between the crystalline materials and energy levels regarding the crystal structural system information.

### 4.3 Energy Decoder

After obtaining the final material-specific energy embedding matrix $\mathbf{E}^{L_3,i}$ of a crystalline material $\mathcal{G}_i$, the DOS value at each energy level $\mathcal{E}_j$, i.e., $\hat{\mathbf{Y}}_j^i$, is given as follows: $\hat{\mathbf{Y}}_j^i = \phi_{pred}(\mathbf{E}_j^{L_3,i})$, where $\phi_{pred} : \mathbb{R}^d \to \mathbb{R}^1$ is a parameterized MLP for predicting DOS. While previous approaches directly predict DOS from the material representation $\mathbf{g}_i$ using a function $\phi_{pred} : \mathbb{R}^d \to \mathbb{R}^M$, DOSTransformer takes a different approach by making pointwise predictions that align with the nature of DOS calculation, i.e., DOS determines the general distribution of states as a function of energy To the best of our knowledge, DOSTransformer is the first work that predicts DOS values in a pointwise manner at each energy level, and we later show in Section 5 that such an approach further enhances the performance of not only DOSTransformer but also existing models.

### 4.4 Model Training and Inference

DOSTransformer is trained to minimize the root mean squared error (RMSE) loss $\mathcal{L}$ between the predicted DOS value $\hat{\mathbf{Y}}_j^i$ and the ground truth DOS value $\mathbf{Y}_j^i$, i.e., $\mathcal{L} = \frac{1}{N \cdot M} \sum_{i=1}^N \sum_{j=1}^M \sqrt{(\hat{\mathbf{Y}}_j^i - \mathbf{Y}_j^i)^2}$. More specifically, DOSTransformer is trained by combining two distinct RMSE losses with balancing term $\beta$, i.e., $\mathcal{L}^{total} = \mathcal{L}^{glob} + \beta \cdot \mathcal{L}^{sys}$, where $\mathcal{L}^{glob}$ and $\mathcal{L}^{sys}$ are obtained by utilizing $\mathbf{E}^{L_3,glob}$ and $\mathbf{E}^{L_3,sys}$, respectively. By doing so, DOSTransformer extracts the relationship information that is shared among the distinct crystal systems and within a crystal system, respectively. For inference, we utilize the model prediction obtained based on $\mathbf{E}^{L_3,sys}$. The overall model architecture is depicted in Figure 2.

## 5 Experiments

### 5.1 Experimental Setup

**Datasets.** We use two datasets to comprehensively evaluate the performance of DOSTransformer, i.e., Phonon DOS and Electron DOS. We use the **Phonon DOS** dataset following the instructions of the official Github repository [3] of a previous work [9]. For **Electron DOS** dataset, we collect crystalline materials and their electron DOS data from Materials Project (MP) website [4]. We provide further detailed preprocessing procedures and statistics on each dataset in Appendix A.

**Methods Compared.** We mainly compare DOSTransformer to recently proposed state-of-the-art method, i.e., E3NN [9], which utilizes the Euclidean network for encoding material representation. We also compare DOSTransformer to simple baseline methods, i.e., MLP and Graph Network [4], which predicts the entire DOS sequence directly from the learned representation of the materials without regarding the energies during training. Moreover, to evaluate the effectiveness of the transformer layer that considers the relationship between the atoms and various energy levels, we integrate energy embeddings into baseline methods for DOS prediction by concatenating the energy embeddings to the material representation. We provide more details on the implementation of DOSTransformer and compared methods in Appendix C and D, respectively.

---

[3] `https://github.com/zhantaochen/phonondos_e3nn`
[4] `https://materialsproject.org/`

Table 1: Overall model performance under in-distribution scenarios (Bulk M. : Bulk Modulus / Band G. : Band Gap / Ferm. E. : Fermi Energy).

| Model | Phonon DOS | | | Electron DOS | | | Physical Properties (MSE) | | |
|---|---|---|---|---|---|---|---|---|---|
| | MSE | MAE | $R^2$ | MSE | MAE | $R^2$ | Bulk M. | Band G. | Ferm. E. |
| **Energy ✗** | | | | | | | | | |
| MLP | 0.309 | 0.106 | 0.576 | 0.347 | 0.130 | 0.487 | 0.695 | 1.597 | 4.650 |
| | (0.016) | (0.003) | (0.016) | (0.015) | (0.003) | (0.029) | (0.031) | (0.171) | (0.202) |
| Graph Network | 0.259 | 0.099 | 0.638 | 0.264 | 0.103 | 0.613 | 0.688 | 1.457 | 3.362 |
| | (0.009) | (0.001) | (0.013) | (0.005) | (0.000) | (0.013) | (0.060) | (0.051) | (0.189) |
| E3NN | 0.210 | 0.077 | 0.705 | 0.296 | 0.109 | 0.552 | 0.504 | 0.896 | 2.925 |
| | (0.004) | (0.001) | (0.007) | (0.005) | (0.001) | (0.013) | (0.033) | (0.093) | (0.111) |
| **Energy ✓** | | | | | | | | | |
| MLP | 0.251 | 0.099 | 0.652 | 0.341 | 0.128 | 0.499 | 0.521 | 1.409 | 4.372 |
| | (0.004) | (0.001) | (0.006) | (0.011) | (0.002) | (0.012) | (0.011) | (0.182) | (0.067) |
| Graph Network | 0.226 | 0.092 | 0.685 | 0.240 | 0.099 | 0.650 | 0.543 | 1.263 | 3.080 |
| | (0.007) | (0.002) | (0.011) | (0.005) | (0.001) | (0.005) | (0.085) | (0.107) | (0.100) |
| E3NN | 0.200 | 0.074 | 0.724 | 0.291 | 0.114 | 0.564 | 0.451 | 1.605 | 3.777 |
| | (0.001) | (0.001) | (0.002) | (0.000) | (0.000) | (0.008) | (0.023) | (0.231) | (0.175) |
| DOSTransformer | **0.191** | **0.071** | **0.733** | **0.221** | **0.089** | **0.679** | **0.427** | **0.461** | **2.337** |
| | (0.003) | (0.002) | (0.004) | (0.006) | (0.001) | (0.006) | (0.024) | (0.019) | (0.094) |

## 5.2  In-Distribution Evaluation

**Evaluation Protocol.** In this section, we evaluate the model performances under in-distribution scenarios. While we evaluate DOSTransformer with given data splits in a previous work [9] for **Phonon DOS** dataset, we randomly split the **Electron DOS** dataset into train/valid/test of 80/10/10%. Note that while all measures are reported in the original scale, we report MSE values for DOS prediction multiplied by a factor of 10 for clear interpretation during all experiments.

**Experimental Results.** In Table 1, we have the following observations: **1)** Comparing the baseline methods that overlook the energy levels for DOS prediction (i.e., Energy ✗) with their counterparts that incorporate both the energy levels and the crystal structure as heterogeneous input modalities through the energy embeddings (i.e., Energy ✓), we find out that using the energy embeddings consistently enhances the model performance. This indicates that making pointwise predictions on each energy level is crucial for DOS prediction as discussed in Section 4.3, which also aligns with the domain knowledge of materials science, i.e., DOS determines the general distribution of states as a function of energy. **2)** However, we observe that E3NN shows a relatively small performance gain compared with other methods after incorporating the energy information. This is because the integration of the energy embedding may confound the model to learn the proper equivariance of the materials. **3)** On the other hand, DOSTransformer outperforms previous methods that do not consider the complex relationships between the atoms in materials and various energy levels via the various attention mechanisms. This again implies that a naive integration of the energy information into previous models cannot fully benefit from the energy information. We also provide qualitative analysis on the obtained DOS in Section 5.5.

**Physical Validity of DOS.** In addition to the accuracy of DOS prediction, we assess the physical validity of the model-predicted DOS by deriving various physical properties, i.e., bulk modulus, band gap, and Fermi energy, of materials, based on the model-predicted DOS. To do so, given the DFT-calculated DOS, which is considered the ground-truth DOS based on which accurate derivation of various physical properties is possible, we train an MLP with a non-linearity in each layer to predict the properties of a crystal structure. Then, based on the obtained MLP weights, we predict the material properties given the model-predicted DOS as the input and calculate the MSE for each material property. We provide further details on the experimental setting in Appendix B. In Table 1, we observe that DOS predicted by DOSTransformer shows the superiority of predicting physical properties of materials, indicating that DOSTransformer not only achieves high accuracy but also produces physically valid predictions.

## 5.3  Out-of-Distribution Evaluation

In this section, we evaluate the model performances under out-of-distribution (OOD) scenarios. It is well recognized that existing DFT calculation-based databases have limitations in terms of

Table 2: Overall model performance in OOD scenarios.

| Model | Out-of-Distribution | | | | | |
| | # Atom Species | | | Crystal System | | |
| | MSE | MAE | $R^2$ | MSE | MAE | $R^2$ |
|---|---|---|---|---|---|---|
| **Energy ✗** | | | | | | |
| MLP | 0.545 (0.005) | 0.161 (0.0001) | 0.250 (0.003) | 0.455 (0.006) | 0.149 (0.002) | 0.437 (0.007) |
| Graph Network | 0.460 (0.015) | 0.144 (0.003) | 0.381 (0.028) | 0.407 (0.004) | 0.132 (0.001) | 0.505 (0.006) |
| E3NN | 0.541 (0.007) | 0.154 (0.001) | 0.231 (0.003) | 0.421 (0.004) | 0.134 (0.001) | 0.483 (0.007) |
| **Energy ✓** | | | | | | |
| MLP | 0.545 (0.005) | 0.159 (0.001) | 0.260 (0.016) | 0.455 (0.004) | 0.149 (0.001) | 0.445 (0.007) |
| Graph Network | 0.455 (0.002) | 0.144 (0.002) | 0.388 (0.010) | 0.384 (0.012) | 0.129 (0.004) | 0.534 (0.014) |
| E3NN | 0.534 (0.013) | 0.152 (0.001) | 0.237 (0.017) | 0.420 (0.007) | 0.133 (0.001) | 0.483 (0.008) |
| DOSTransformer | **0.454** (0.008) | **0.136** (0.001) | **0.399** (0.017) | **0.373** (0.006) | **0.122** (0.001) | **0.552** (0.008) |

Table 3: Fine-tuning on OOD systems.

| Model | MSE | MAE | $R^2$ |
|---|---|---|---|
| **Energy ✗** | | | |
| MLP | 0.401 (0.017) | 0.137 (0.003) | 0.510 (0.021) |
| Graph Network | 0.394 (0.007) | 0.131 (0.002) | 0.519 (0.010) |
| E3NN | 0.414 (0.006) | 0.133 (0.001) | 0.490 (0.007) |
| **Energy ✓** | | | |
| MLP | 0.394 (0.008) | 0.136 (0.002) | 0.519 (0.009) |
| Graph Network | 0.382 (0.003) | 0.130 (0.000) | 0.533 (0.002) |
| E3NN | 0.417 (0.004) | 0.133 (0.001) | 0.487 (0.007) |
| **DOSTransformer** | | | |
| All | 0.365 (0.005) | **0.122** (0.001) | 0.558 (0.008) |
| Only Prompt | **0.355** (0.008) | **0.122** (0.001) | **0.570** (0.010) |

coverage, as they often focus on specific types of materials or structural archetypes, resulting in biased distributions [13, 30, 20]. Therefore, it is essential to evaluate the model's performance in OOD scenarios to assess its real-world applicability and generalizability beyond the limitations of the available databases [35].

**Evaluation Protocol.** To do so, we evaluate the model performance on the crystal structures that 1) contain a different number of atom species with the training set (i.e., # Atom species), and 2) belong to different crystal systems that were not included in the training set (i.e., Crystal System). In both scenarios, training data primarily consist of relatively simple structures compared to those in the test data. We provide further details on data split and evaluation in Appendix B.

**Experimental Results.** In Table 2, we again observe that utilizing energy embeddings consistently brings performance gain to all baseline models. Especially, Graph Network significantly benefits from energy embeddings compared to in-distribution scenarios (See Table 1). We attribute this to the inherent restrictive inductive biases in Graph Network [4], i.e., Graph Network heavily relies on the structural information of materials. Specifically, as the structure of the materials totally varies in the OOD scenarios, which makes the prediction of the whole DOS sequence challenging, incorporating the energy embeddings becomes especially helpful in the OOD scenarios for Graph Network. Moreover, DOSTransformer also alleviates the restrictive inductive bias of Graph Networks by elaborately modeling the complex relationships between energies and materials. It is worth noting that similar observations have been made by comparing Convolutional Neural Networks (CNNs) and Vision Transformers (ViTs) in the computer vision domain [1, 66]. In conclusion, we argue that incorporating energy information during model training is crucial not only for in-distribution scenarios but also for OOD scenarios, demonstrating the applicability of DOSTransformer in real-world applications. For a more detailed analysis, please refer to Appendix E.1.

**Fine-tuning on Few Materials from Complex Crystal Systems.** In addition to evaluating the performance under the OOD scenarios, we verify how well the models can make predictions for materials from complex crystal systems with only few training samples, which is a practical situation in reality. For this, we sampled a small subset of of materials (i.e., 10%) from the test set used in OOD scenarios of crystal systems, and fine-tune the models that are already trained on the training set. As for DOSTransformer, we fine-tune the model using two different approaches: i) fine-tuning all model parameters (referred to as "All"), and ii) tuning only the parameters of the prompts and the energy decoder while keeping all other model parameters frozen (referred to as "Only Prompt"). In Table 3, we have the following observations: **1)** As we expected, the additional fine-tuning step achieves performance gain for all models, while it was marginal due to a limited number of materials used for fine-tuning. **2)** Only fine-tuning the prompts of DOSTransformer achieved more performance gain compared to fine-tuning the whole model. This is because while fine-tuning the whole model on a small subset of materials may easily incur overfitting, fine-tuning only prompts enables the model to additionally learn from few new samples while maintaining the knowledge obtained previously. In conclusion, we believe that the proposed prompt tuning approach can have a significant impact in the

field of materials science beyond simple DOS prediction, where the existing databases exhibit a bias towards dominant types of materials [13, 30, 20]. We further explore different proportions beyond the 10% of the materials used for fine-tuning, as detailed in the Appendix E.3.

## 5.4 Model Analysis

**Ablation Studies.** To verify the benefit of each component of DOS-Transformer, we conduct ablation studies under in-distribution scenarios in Figure 3. We observe that utilizing only either one of the global and system losses deteriorates the model performance. This is because jointly optimizing the losses incentivizes the model to extract the relational information that is shared across the crystal systems and within a crystal system. On the other hand, it is worth noting that DOSTransformer with only one of the losses still outperforms baseline models shown in Table 1, indicating that modeling complex relationships between materials and various energies is crucial in DOS prediction.

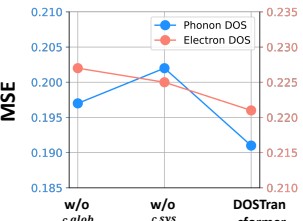

Figure 3: Ablation studies.

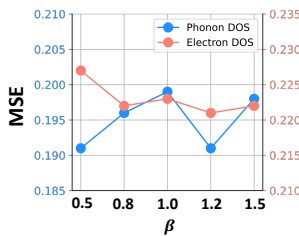

Figure 4: Sensitivity analysis.

**Sensitivity Analysis.** To verify the robustness of DOSTransformer, we measure the model performance by varying $\beta$, which is introduced to control the effect of the crystal system loss, i.e. $\mathcal{L}^{sys}$, in Section 4.4. In Figure 4, DOSTransformer shows robustness over various levels of $\beta$, consistently outperforming baseline models in Table 1. This verifies that DOSTransformer can be easily trained without an expensive hyperparameter tuning, further demonstrating the practicality of DOSTransformer.

**Crystal System Prompts.** As discussed in Section 4.2, it is not trivial to effectively incorporate the crystal system information into the model. To evaluate the effectiveness of our proposed prompt-based approach, we explore different approaches for incorporating the crystal system information into the model. We make modifications to determine where and how to inject this information. In terms of the location, we consider two options: injecting the information into the input atoms (i.e., Atom Feat.) and injecting it before the self-attention layers (i.e., Before SA). Regarding the method

Table 4: Comparing various crystal system information injection approaches.

| | | Method | | Electron DOS | |
|---|---|---|---|---|---|
| | | One-hot | Prompt | MSE | MAE |
| **Location** | Atom Feat. | ✓ | | 0.222 (0.004) | 0.089 (0.001) |
| | | | ✓ | 0.227 (0.005) | 0.090 (0.001) |
| | Before SA | ✓ | | 0.226 (0.005) | 0.090 (0.001) |
| | | | ✓ | **0.221** (0.006) | **0.089** (0.001) |

of injection, we explore two strategies: one-hot encoding of crystal systems and the use of learnable crystal system prompts. In Table 4, we have following observations: **1)** Naively incorporating prompts into atom feature even perform worse than the model without using crystal system (see "*w/o $\mathcal{L}^{sys}$*" in Figure 3), demonstrating the importance of an elaborate design choice for injecting crystal system information. **2)** We observe that our approach, i.e. Before SA and prompt, outperforms all other possible choices, indicating that DOSTransformer successfully incorporates crystal system information during training. We also examine the benefits of injecting crystal system information into baseline methods in Appendix E.2.

## 5.5 Qualitative Analysis

In this section, we provide a qualitative analysis of the predicted DOS by mainly comparing it to the DFT-calculated (i.e., Ground Truth) DOS and our main baseline (i.e., E3NN). In Figure 5 (a), which represents the predicted DOS of materials not containing transitional metals, both E3NN and DOSTransformer successfully capture the overall trend of the DOS for several materials (e.g., mp-982366, mp-1009129, and mp-16378). However, DOSTransformer shows a much more precise prediction that closely aligns with the ground truth DOS, providing even more useful information beyond the shape of DOS. For example, peak points represent regions of high density and are likely to be strongly influenced when materials undergo changes in property, and thus represents the probabilistically important energy regions of the materials in the process of material discovery. Notably, our model better captures the peak points in the ground truth DOS compared to E3NN, demonstrating the applicability of DOSTransformer-predicted DOS for material discovery process.

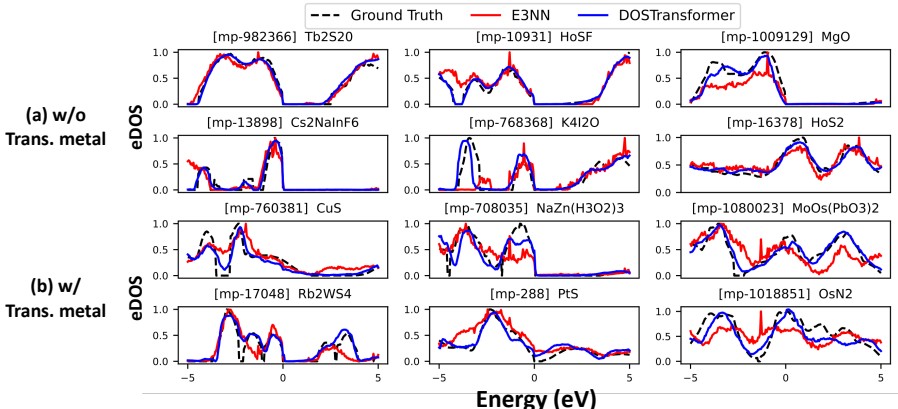

Figure 5: Qualitative Analysis.

On the other hand, Figure 5 (b) shows the DOS prediction for materials containing transition materials. Although DOSTransformer provides more reliable prediction, we observe that the prediction errors of both models get larger compared to the materials that do not contain transition metals shown in Figure 5 (a). This can be attributed to the inherent complexity of physical properties in materials containing transition metals, as discussed in Section 6. Therefore, for our future work, we plan to design expert models in which each expert is responsible for materials with and without transition metals, to achieve more refined and accurate predictions of the DOS. This approach would enable a more comprehensive and elaborate analysis of the DOS in different material compositions.

# 6 Limitations & Future Work

It is widely known that materials containing transitional metals inherently have complex physical properties compared to other materials. As a result, the density functionals that yield optimal performance in DFT calculations differ between materials with and without transitional metals [12, 46]. However, despite the distinction, most ML-based DOS prediction research, including DOSTransformer, have attempted to encode the properties of both material types into a single model [9, 17], which may provide conflicting signals to the model. Therefore, as a future work, we plan to address this issue by developing specialized approaches that work well on both material types, e.g., utilizing expert models [41] in which an expert is assigned to each material type and later combined to perform well on both material types. We expect this approach to result in the robust and reliable prediction of DOS for both types of materials, enhancing the potential impact in the materials science field and broadening our understanding of material properties.

# 7 Conclusion

In this paper, we propose DOSTransformer, which predicts DOS of crystalline materials with various energy levels by following the nature of DOS calculation. Specifically, DOSTransformer associates the heterogeneous information from materials and energy levels with cross-attention and self-attention layers. Moreover, the crystal system prompts are introduced to effectively train the model to learn the relational information that is shared across all the crystal systems and within a crystal system. By doing so, DOSTransformer outperforms previous works in predicting two types of DOS, i.e., phonon DOS and electron DOS, in various scenarios, i.e., in-distribution scenarios and out-of-distribution scenarios. Extensive experiments verify that incorporating energy information is crucial in predicting the DOS of a crystal structure for real-world application and DOSTransformer effectively utilizes the crystal structural information with crystal system prompts.

## Acknowledgements

This work was supported by the core KRICT project from the Korea Research Institute of Chemical Technology (SI2051-10).

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

# Supplementary Material for
# Density of States Prediction of Crystalline Mateirals via Prompt-guided Multi-Modal Transformer

## A  Datasets

In this section, we provide further details on the dataset used for experiments.

### A.1  Phonon DOS

We use the **Phonon DOS** dataset following the instructions of the official Github repository[5] of a previous work [9]. This dataset contains 1,522 crystalline materials whose phonon DOS is calculated from density functional perturbation theory (DFPT) by a previous work [42]. Since the provided dataset does not contain crystal system information, we additionally collect the information based on the Materials Project (MP) website [4] on the given each material's unique ID (MP-id).

### A.2  Electron DOS

We also use **Electron DOS** dataset that contains 38,889 crystalline materials. The Electron DOS dataset consists of the materials and their electron DOS information that is collected from the MP website [4]. Among the collected data, we exclude the materials that are tagged to include magnetism because the DOS of magnetism materials is not accurate to be directly used for training machine learning models [23]. We consider an energy grid of 201 points ranging from $-5$ to 5 eV with respect to the band edges with 50 meV intervals and the Fermi energy is all set to 0 eV on this energy grid. Moreover, we normalize the DOS of each material to be in the range between 0 and 1. That is, the maximum and minimum value for each DOS is 1 and 0, respectively, for all materials. Moreover, we smooth the DOS values with the Savitzky-Golay filter with the window size of 17 and polyorder of 1 using scipy library following a previous work [9].

### A.3  Data Statistics of Electron DOS dataset in OOD scenarios

As described in the main manuscript, we further evaluate the model performance in two out-of-distribution scenarios: **Scenario 1**: regarding the number of atom species, and **Scenario 2**: regarding the crystal systems. We provide detailed statistics of the number of crystalline materials for each scenario in Table 5 and Table 6.

Table 5: The number of crystals according to the number of atom species (Scenario 1).

|  | Unary (1) | Binary (2) | Ternary (3) | Quaternary (4) | Quinary (5) | Senary (6) | Septenary (7) | Total |
|---|---|---|---|---|---|---|---|---|
| # Materials | 386 | 9,034 | 21,794 | 5,612 | 1,750 | 279 | 34 | 38,889 |

Table 6: The number of crystals according to different crystal systems (Scenario 2).

|  | Cubic | Hexagonal | Tetragonal | Trigonal | Orthorhombic | Monoclinic | Triclinic | Total |
|---|---|---|---|---|---|---|---|---|
| # Materials | 8,385 | 3,983 | 5,772 | 3,964 | 8,108 | 6,576 | 2,101 | 38,889 |

---

[5]`https://github.com/zhantaochen/phonondos_e3nn`

## B  Evaluation Protocol

**Phonon DOS.** As described in the main manuscript, we evaluate the model performance based on the data splits given in a previous work [9].

**Electron DOS.** On the other hand, for the Electron DOS dataset, we use different dataset split strategies for each scenario. For the in-distribution setting, we randomly split the dataset into train/valid/test of 80/10/10%. On the other hand, for the out-of-distribution setting, we split the dataset regarding the structure of the crystals. For both scenarios, we generate training sets with simple crystal structures and a valid/test set with more complex crystal structures, because it is crucial to transfer the knowledge obtained from simple crystal structures to that from complex structures in real-world materials science. More specifically, in the scenario 1 (different number of atom species, i.e., # Atom species in Table 2), we use Binary and Ternary materials as training data and Unary, Quaternary, and Quinary materials as valid and test data. In the case of Unary, we exclude it from training data despite its simplicity due to the observed difficulty of the structure, as will be discussed in Section E.1. In the scenario 2 (different crystal systems, i.e., Crystal System in Table 2), we use Cubic, Hexagonal, Tetragonal, Trigonal, and Orthorhombic crystal systems as training set and Monoclinic and Triclinic as valid and test set. In this scenario, where no prompt is available for unseen crystal systems, we employ the mean-pooled representations of the trained prompts during testing, i.e., for the Monoclinic and Triclinic crystal systems. Please refer to Table 5 and Table 6 for detailed statistics of crystals in each scenario.

**Physical Properties.** In addition to evaluating the accuracy of the model's predictions of the DOS, it is crucial to assess the physical meaningfulness of the predicted DOS for real-world applications. To assess the physical meaningfulness of the predicted DOS, we utilize the predicted DOS to estimate a range of important material properties. Specifically, we evaluate three materials' properties: the bulk modulus for phonon DOS, and the band gap and Fermi energy for electron DOS (Table 1).

Bulk Modulus [6] is a thermodynamic quantity measuring the resistance of a substance to compression. It provides a measure of the material's ability to withstand changes in volume under applied pressure. In the context of elastic properties, the bulk modulus serves as a descriptor, as it indicates how well a material can recover its original volume after being subjected to compression.

Another property we focus on is the Band Gap [7], which refers to the energy range in a material where no electronic states exist. It represents the energy difference between the top of the valence band and the bottom of the conduction band in insulators and semiconductors. Functional inorganic materials, such as those used in applications like LEDs, transistors, photovoltaics, or scintillators, require a comprehensive understanding of their band gap [69]. By accurately predicting the band gap based on DOS, we can accelerate the development of new materials for a wide range of applications.

Additionally, we predict the Fermi Energy [8], which represents the highest energy level occupied by electrons at absolute zero temperature (0K). It can be used to determine the electrical and thermal characteristics of materials.

## C  Implementation Details

In this section, we provide implementation details of DOSTransformer.

**Graph Neural Networks.** Our graph neural networks consist of two parts, i.e., encoder and processor. Encoder learns the initial representation of atoms and bonds, while the processor learns to pass the messages across the crystal structure. More formally, given an atom $v_i$ and the bond $e_{ij}$ between atom $v_i$ and $v_j$, node encoder $\phi_{node}$ and edge encoder $\phi_{edge}$ outputs initial representations of atom $v_i$ and bond $e_{ij}$ as follows:

$$\mathbf{h}_i^0 = \phi_{node}(\mathbf{X}_i), \quad \mathbf{b}_{ij}^0 = \phi_{edge}(\mathbf{B}_{ij}), \tag{4}$$

where $\mathbf{X}$ is the atom feature matrix whose $i$-th row indicates the input feature of atom $v_i$, $\mathbf{B} \in \mathbb{R}^{n \times n \times F_e}$ is the bond feature tensor with $F_e$ features for each bond. With the initial representations

---

[6] https://en.wikipedia.org/wiki/Bulk_modulus
[7] https://en.wikipedia.org/wiki/Band_gap
[8] https://en.wikipedia.org/wiki/Fermi_energy

of atoms and bonds, the processor learns to pass messages across the crystal structure and update atoms and bonds representations as follows:

$$\mathbf{b}_{ij}^{l+1} = \psi_{edge}^l(\mathbf{h}_i^l, \mathbf{h}_j^l, \mathbf{b}_{ij}^l), \quad \mathbf{h}_i^{l+1} = \psi_{node}^l(\mathbf{h}_i^l, \sum_{j \in \mathcal{N}(i)} \mathbf{b}_{ij}^{l+1}), \tag{5}$$

where $\mathcal{N}(i)$ is the neighboring atoms of atom $v_i$, $\psi$ is a two-layer MLP with non-linearity, and $l = 0, \ldots, L'$. Note that $\mathbf{h}_i^{L'}$ is equivalent to the $i$-th row of the atom embedding matrix $\mathbf{H}$ in Equation 1.

**Model Training.** In all our experiments, we use the AdamW optimizer for model optimization. For all the tasks, we train the model for 1,000 epochs with early stopping applied if the best validation loss does not change for 50 consecutive epochs.

**Hyperparameter Tuning.** Detailed hyperparameter specifications are given in Table 7. For the hyperparameters in DOSTransformer, we tune them in certain ranges as follows: number of message passing layers in GNN $L'$ in {2, 3, 4}, number of cross-attention layers $L_1$, $L_3$ in {2, 3, 4}, number of self-attention layers $L_2$ in {2, 3, 4}, hidden dimension $d$ in {64, 128, 256}, learning rate $\eta$ in {0.0001, 0.0005, 0.001}, and batch size $B$ in {1, 4, 8}. We use the sum pooling to obtain the crystalline material $i$'s representation, i.e., $\mathbf{g}_i$. We report the test performance when the performance on the validation set gives the best result.

Table 7: Hyperparameter specifications of DOSTransformer.

| Hyperparameters | In-Distribution | | Out-of-Distribution | |
|---|---|---|---|---|
| | Phonon DOS | Electron DOS | # Atom Species | Crystal Systems |
| # Message Passing Layers ($L'$) | 3 | 3 | 3 | 3 |
| # Cross-Attention Layers ($L_1$) | 2 | 2 | 2 | 2 |
| # Self-Attention Layers ($L_2$) | 2 | 2 | 2 | 2 |
| # Cross-Attention Layers ($L_3$) | 2 | 2 | 2 | 2 |
| Hidden Dim. ($d$) | 256 | 256 | 256 | 256 |
| Learning Rate ($\eta$) | 0.0001 | 0.0001 | 0.0001 | 0.0001 |
| Batch Size ($B$) | 1 | 8 | 8 | 8 |

# D   Methods Compared

In this section, we provide further details on the methods that are compared with DOSTransformer in our experiments.

**MLP.** We first encode the atoms in a crystalline material with an MLP. Then, we obtain the representation of material $i$, i.e., $\mathbf{g}_i$, by sum pooling the representations of its constituent atoms. With the material representation, we predict DOS with an MLP predictor $\phi'$, i.e., $\hat{\mathbf{Y}}^i = \phi'(\mathbf{g}_i)$, where $\phi' : \mathbb{R}^d \rightarrow \mathbb{R}^{201}$.

On the other hand, when we incorporate energy embeddings into the MLP, we predict DOS for each energy $j$ with a learnable energy embedding $\mathbf{E}_j^0$ and obtained material representation $\mathbf{g}_i$, i.e., $\hat{\mathbf{Y}}_j^i = \phi(\mathbf{E}_j^0 || \mathbf{g}_i)$, where $\phi : \mathbb{R}^{2d} \rightarrow \mathbb{R}^1$ is a parameterized MLP.

**Graph Network.** We first encode the atoms in a crystalline material with a graph network [4]. As done for MLP, we obtain the representation of material $i$, i.e., $\mathbf{g}_i$, by sum pooling the representations of its constituent atoms. With the material representation, we predict the DOS with an MLP predictor,

i.e., $\hat{\mathbf{Y}}^i = \phi'(\mathbf{g}_i)$, where $\phi' : \mathbb{R}^d \to \mathbb{R}^{201}$. Note that the only difference with MLP is that the atom representations are obtained through the message passing scheme. We also compare the vanilla graph network that incorporates the energy information as we have done in MLP.

**E3NN.** For E3NN [9], we use the official code published by the authors[9], which implements equivariant neural networks with E3NN python library[10]. By learning the equivariance, the model can generate high-quality representations with a small number of training materials. After obtaining the crystalline material representation $\mathbf{g}_i$, all other procedures have been done in the same manner with other baseline models, i.e., MLP and Graph Network.

# E  Additional Experiments

## E.1  Model Performance Analysis on Out-of-Distribution Scenarios

In this section, we conduct a comprehensive analysis of the model's predictions in the out-of-distribution scenarios presented in Table 2. In Table 8, we evaluate the performance of the model for each type of material, providing detailed insights into its predictive capabilities. We have following observations: **1)** We observe that DOSTransformer consistently outperforms in both out-of-distribution scenarios, which demonstrates the superiority of DOSTransformer. **2)** The performance of all the compared models generally degrades as the crystal structure gets more complex. That is, models perform worse in Quinary crystals than in Quarternary crystals, and worse in Triclinic crystals than in Monoclinic crystals. **3)** On the other hand, it is not the case in Unary crystal. This is because in Unary crystal only one type of atom repeatedly appears in the crystal structure, which cannot give enough information to the model. However, DOSTransformer also makes comparably accurate predictions in the Unary materials by modeling the complex relationship between the atoms and various energy levels.

Table 8: Model performance in Out-of-Distribution scenarios.

| Model | # Atom Species | | | | | | Crystal System | | | |
|---|---|---|---|---|---|---|---|---|---|---|
| | Unary | | Quarternary | | Quinary | | Monoclinic | | Triclinic | |
| | MSE | MAE | MSE | MAE | MSE | MAE | MSE | MAE | MSE | MAE |
| **Energy ✗** | | | | | | | | | | |
| MLP | 0.578 | 0.180 | 0.500 | 0.153 | 0.673 | 0.182 | 0.370 | 0.132 | 0.470 | 0.146 |
| | (0.005) | (0.001) | (0.003) | (0.001) | (0.004) | (0.002) | (0.017) | (0.003) | (0.021) | (0.003) |
| Graph Network | 0.485 | 0.165 | 0.443 | 0.141 | 0.620 | 0.170 | 0.376 | 0.127 | 0.504 | 0.147 |
| | (0.013) | (0.003) | (0.003) | (0.001) | (0.007) | (0.003) | (0.003) | (0.001) | (0.007) | (0.001) |
| E3NN | 0.565 | 0.167 | 0.486 | 0.145 | 0.708 | 0.179 | 0.393 | 0.129 | 0.510 | 0.148 |
| | (0.025) | (0.002) | (0.001) | (0.000) | (0.013) | (0.002) | (0.003) | (0.001) | (0.006) | (0.002) |
| **Energy ✓** | | | | | | | | | | |
| MLP | 0.597 | 0.183 | 0.498 | 0.151 | 0.684 | 0.178 | 0.367 | 0.130 | 0.468 | 0.144 |
| | (0.034) | (0.005) | (0.003) | (0.002) | (0.015) | (0.002) | (0.012) | (0.002) | (0.020) | (0.003) |
| Graph Network | 0.471 | 0.160 | 0.416 | 0.137 | 0.571 | 0.165 | 0.359 | 0.125 | 0.476 | 0.144 |
| | (0.028) | (0.002) | (0.002) | (0.002) | (0.005) | (0.002) | (0.010) | (0.004) | (0.008) | (0.004) |
| E3NN | 0.567 | 0.166 | 0.481 | 0.143 | 0.689 | 0.175 | 0.393 | 0.128 | 0.516 | 0.147 |
| | (0.021) | (0.003) | (0.008) | (0.000) | (0.020) | (0.002) | (0.006) | (0.001) | (0.006) | (0.001) |
| DOSTransformer | **0.417** | **0.145** | **0.413** | **0.128** | **0.570** | **0.157** | **0.343** | **0.117** | **0.466** | **0.137** |
| | (0.012) | (0.003) | (0.010) | (0.002) | (0.010) | (0.001) | (0.003) | (0.001) | (0.007) | (0.001) |

## E.2  Injecting Crystal System Information to Baseline Methods

In this section, we adopt our prompt-based crystal system information injection procedure to the baseline methods. We examine two approaches for injecting the information: 1) injecting the information into the input atoms (i.e., Position 1), and 2) injecting it before the DOS prediction layer (i.e., Position 2). In Table 9, we have the following observations: **1)** Compared to Table 1, all baseline models benefit from using crystal system information. This demonstrates the importance of

---

[9] https://github.com/ninarina12/phononDoS_tutorial
[10] https://docs.e3nn.org/en/latest/index.html

utilizing crystal structural systems information, which has been overlooked in previous works. **2)** However, DOSTransformer still outperforms all baseline methods with crystal system information (See DOSTransformer in Table 1), verifying the importance of an elaborate design of crystal system injection procedure. To be more specific, we notice a relatively significant performance gap between DOSTransformer and the best baseline model in Electron DOS, which comprises a broader range of crystalline materials than Phonon DOS. This finding highlights the importance of an intricate crystal system injection procedure when striving to learn the DOS of diverse crystalline materials.

Table 9: Baseline model performance with crystal structural system prompts.

| Model | Phonon DOS | | | | Electron DOS | | | |
|---|---|---|---|---|---|---|---|---|
| | Position 1 | | Position 2 | | Position 1 | | Position 2 | |
| | MSE | MAE | MSE | MAE | MSE | MAE | MSE | MAE |
| **Energy ✗** | | | | | | | | |
| MLP | 0.308 | 0.105 | 0.324 | 0.109 | 0.343 | 0.128 | 0.340 | 0.127 |
| | (0.013) | (0.001) | (0.012) | (0.002) | (0.008) | (0.001) | (0.009) | (0.0031 |
| Graph Network | 0.260 | 0.097 | 0.244 | 0.095 | 0.260 | 0.102 | 0.260 | 0.103 |
| | (0.008) | (0.000) | (0.023) | (0.001) | (0.011) | (0.001) | (0.006) | (0.002) |
| E3NN | 0.210 | 0.077 | 0.209 | 0.079 | 0.292 | 0.108 | 0.299 | 0.110 |
| | (0.007) | (0.002) | (0.010) | (0.001) | (0.007) | (0.001) | (0.002) | (0.001) |
| **Energy ✓** | | | | | | | | |
| MLP | 0.251 | 0.100 | 0.245 | 0.098 | 0.332 | 0.125 | 0.336 | 0.127 |
| | (0.000) | (0.000) | (0.004) | (0.001) | (0.003) | (0.001) | (0.007) | (0.002) |
| Graph Network | 0.230 | 0.094 | 0.224 | 0.091 | 0.234 | 0.097 | 0.230 | 0.093 |
| | (0.009) | (0.002) | (0.009) | (0.002) | (0.001) | (0.004) | (0.009) | (0.002) |
| E3NN | 0.194 | 0.073 | 0.190 | 0.073 | 0.286 | 0.108 | 0.290 | 0.109 |
| | (0.004) | (0.000) | (0.000) | (0.001) | (0.002) | (0.000) | (0.003) | (0.000) |

### E.3 Various Training Data Ratio for Fine-Tuning

In this section, we additionally provide experimental results on various ratios of training data for fine-tuning in Table 3. That is, instead of sampling 10% of training data from the test set used in OOD scenarios in Section 5.3, we try various sampling ratios, i.e., 5%, 10%, 15%, and 20%, from the test set. We have the following observations: **1)** We notice a significant performance disparity between the "Only Prompt" and "All" approaches, particularly when the training dataset is limited. This phenomenon can be attributed to the challenge of overfitting when fine-tuning the entire model on a small subset of materials, as discussed in Section 5.3. **2)** Conversely, as the training data increases, we observe that the difference in performance between the "Only Prompt" and "All" methods diminishes. This is due to the enough amount of training data allowing for the adjustment of

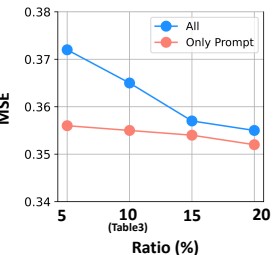

Figure 6: Various training data ratios for fine-tuning.

all model parameters, consistent with the analysis discussed in Section 5.3. However, it is important to note that the existing DFT calculation-based databases suffer from a highly biased distribution, which limits their coverage of different materials. This limitation emphasizes the significance of achieving good performance even with a small subset of training data. Therefore, we argue that the application of prompt tuning enhances the real-world applicability of DOSTransformer.

### E.4 Comparison with Crystal Neural Network

We conducted additional experiments on the state-of-the-art crystal neural networks, i.e., ALIGNN [10] and Matformer [63], by adjusting the output dimensions of these models (51 for Phonon DOS and 201 for Electron DOS) in Table 10. Although both ALIGNN and Matformer exhibited commendable performance, we observe that DOSTransformer consistently surpassed them significantly. These observations again demonstrate the importance of considering energy levels for accurate DOS prediction.

Table 10: Comparison to crystal neural networks.

| Model | Phonon DOS | | | Electron DOS | | |
|---|---|---|---|---|---|---|
| | MSE | MAE | $R^2$ | MSE | MAE | $R^2$ |
| ALIGNN | 0.204 | 0.085 | 0.717 | 0.270 | 0.108 | 0.600 |
| | (0.005) | (0.001) | (0.007) | (0.004) | (0.001) | (0.001) |
| Matformer | 0.198 | 0.084 | 0.724 | 0.268 | 0.104 | 0.601 |
| | (0.008) | (0.002) | (0.013) | (0.004) | (0.001) | (0.002) |
| DOSTransformer | **0.191** | **0.071** | **0.733** | **0.221** | **0.089** | **0.679** |
| | (0.003) | (0.002) | (0.004) | (0.006) | (0.001) | (0.006) |

## E.5   Model Training and Inference Time

In this section, to verify the efficiency of DOSTransformer, we compare the training and inference time of the baseline methods in Table 11. We observe that DOSTransformer requires a longer training time per epoch on the Phonon DOS dataset compared to E3NN, which can be attributed to the two forward passes (i.e., system and global energy embeddings) during the training procedure. However, when it comes to the Electron DOS dataset, DOSTransformer demonstrates a shorter training time per epoch compared to E3NN. This is because the Electron DOS dataset has complex crystal structures, requiring more time for E3NN to learn equivariant representations. Furthermore, in terms of inference time, DOSTransformer demonstrates significantly faster computation per epoch compared to E3NN, particularly on the Electron DOS dataset. This is because we only utilize system prediction without global prediction during inference. As many predictive ML models are used for high-throughput screening in material discovery, inference time is a critical factor for ML models in materials science, demonstrating the practicality of DOSTransformer in real-world applications.

Table 11: Training and inference time per epoch for each dataset (sec/epoch).

| Model | Training | | Inference | |
|---|---|---|---|---|
| | Phonon DOS | Electron DOS | Phonon DOS | Electron DOS |
| **Energy ✗** | | | | |
| MLP | 4.10 | 23.52 | 1.51 | 3.00 |
| Graph Network | 16.17 | 59.74 | 1.95 | 3.88 |
| E3NN | 21.21 | 141.02 | 3.72 | 9.49 |
| **Energy ✓** | | | | |
| MLP | 4.67 | 27.88 | 1.66 | 3.10 |
| Graph Network | 17.45 | 66.83 | 2.16 | 4.28 |
| E3NN | 24.12 | 152.80 | 3.92 | 10.21 |
| DOSTransformer | 39.17 | 145.85 | 2.98 | 5.99 |

## F   Broader Impacts

**Potential Positive Scientific Impacts.**   In this work, we propose DOSTransformer, which is the first work that considers various energy levels during DOS prediction and introduces prompts for crystal structural system, demonstrating its applicability in real-world scenarios. For example, transferring the knowledge obtained from simple structured materials to complex structured materials is crucial because DFT calculation-based databases cover limited types of materials or structural archetypes. Therefore, we believe DOSTransformer has broad impacts on various fields of materials science.

**Potential Negative Societal Impacts.**   This work explores the automation process for materials science without wet lab experiments. However, it is important to acknowledge that in the industry, there are skilled professionals dedicated to conducting such experiments for materials science.

Therefore, it is important to proactively address these concerns by encouraging collaboration between automated methods and human experts.

# G  Pseudo Code

Algorithm 1 shows the pseudocode of DOSTransformer.

---
**Algorithm 1:** Pseudocode of DOSTransformer.

---
**Input:** An input crystalline material $\mathcal{G} = (\mathbf{X}, \mathbf{A})$, Ground truth DOS $\mathbf{Y}$, Number of attention layers $L_1, L_2, L_3$, Initialized energy embeddings $\mathbf{E}$, Initialized crystal system prompts $\mathbf{P}$.

1   $\mathbf{H} \leftarrow \text{GNN}(\mathbf{X}, \mathbf{A})$
2   $\mathbf{E}^{L_1} \leftarrow \text{Cross-Attention}(\mathbf{H}, \mathbf{E}, L_1)$
3   $\mathbf{g} \leftarrow \text{Sum Pooling}(\mathbf{H})$

4   $\mathbf{E}^{glob} \leftarrow (\mathbf{E}^{L_1} || \mathbf{g})$
5   $\tilde{\mathbf{E}}^{0,glob} \leftarrow \phi_1(\mathbf{E}^{glob})$
6   $\tilde{\mathbf{E}}^{L_2,glob} \leftarrow \text{Self-Attention}(\tilde{\mathbf{E}}^{0,glob}, L_2)$   `// Global Self-Attention`
7   $\mathbf{E}^{L_3,glob} \leftarrow \text{Cross-Attention}(\mathbf{H}, \tilde{\mathbf{E}}^{L_2,glob}, L_3)$
8   $\hat{\mathbf{Y}} \leftarrow \phi_{pred}(\mathbf{E}^{L_3,glob})$
9   $\mathcal{L}^{glob} \leftarrow \text{RMSE}(\hat{\mathbf{Y}}, \mathbf{Y})$

10   $\mathbf{E}^{sys} \leftarrow (\mathbf{E}^{L_1} || \mathbf{g} || \mathbf{P})$
11   $\tilde{\mathbf{E}}^{0,sys} \leftarrow \phi_2(\mathbf{E}^{sys})$
12   $\tilde{\mathbf{E}}^{L_2,sys} \leftarrow \text{Self-Attention}(\tilde{\mathbf{E}}^{0,sys}, L_2)$   `// System Self-Attention`
13   $\mathbf{E}^{L_3,sys} \leftarrow \text{Cross-Attention}(\mathbf{H}, \tilde{\mathbf{E}}^{L_2,sys}, L_3)$
14   $\hat{\mathbf{Y}} \leftarrow \phi_{pred}(\mathbf{E}^{L_3,sys})$
15   $\mathcal{L}^{sys} \leftarrow \text{RMSE}(\hat{\mathbf{Y}}, \mathbf{Y})$

16   $\mathcal{L}^{total} \leftarrow \mathcal{L}^{glob} + \beta \cdot \mathcal{L}^{sys}$   `// Calculate total loss`

17   **Function** Cross-Attention($\mathbf{H}, \mathbf{E}^0, L$)**:**
18     **for** $l = 1, 2, \ldots, L$ **do**
19       $\mathbf{E}^l \leftarrow \text{Softmax}(\frac{\mathbf{E}^{l-1}\mathbf{H}^\top}{\mathbf{H}})$
20     **end**
21     **return** $\mathbf{E}^L$

22   **Function** Self-Attention($\tilde{\mathbf{E}}^0, L$)**:**
23     **for** $p = 1, 2, \ldots, L$ **do**
24       $\tilde{\mathbf{E}}^p \leftarrow \text{Softmax}(\frac{\tilde{\mathbf{E}}^{p-1}\tilde{\mathbf{E}}^{p-1\top}}{\tilde{\mathbf{E}}^{p-1}})$
25     **end**
26     **return** $\tilde{\mathbf{E}}^L$

---

