# OpenReview forum: "Density of States Prediction of Crystalline Materials via Prompt-guided Multi-Modal Transformer"
_NeurIPS.cc/2023/Conference — NeurIPS 2023 poster_

### Official Review · Reviewer_pieM · 2023-06-29

**Soundness:** 3 good
**Presentation:** 3 good
**Contribution:** 2 fair
**Rating:** 6
**Confidence:** 4

**Summary:**

This paper propose DOSTransformer, a transformer architecture for the task of DOS prediction, that takes energy levels as input instead of predicting a list of energies for different energy levels. The performances of this proposed DOSTransformer is better than MLP, GNN, and E3NN instances they implemented.

**Strengths:**

- The proposed method is the first one to take energy levels as input instead of directly predict a list of energies for different energy levels.
- Writing: clear and easy to follow. Figures are informative.

**Weaknesses:**

- Novelty is limited, core ablation study is missing: the major novelty beyond previous works is taking energy levels as input instead of directly predict a list of energies for different energy levels. To do this, they proposed the multi-modal module that considers energy levels. But this point is not well supported by the performance ablation studies of E3NN. Also, the ablation study for their proposed model about this point (just directly predict a list of energies) is missing.
- Missing comparisons with previous crystal neural networks. Previous crystal neural networks including CGCNN, MEGNET, Matformer, ALIGNN, M3GNet can all directly be used for this DOS task, by just changing the output dimensionality from 1 (single scalar value prediction) to M. Just ignoring all these methods is not convincing. Comparing with these methods (at least one or two current SOTA methods) can better demonstrate the importance of taking energy levels as conditions instead of predicting a list of energies.
- Missing implementation details and what level of irreducible representations is used in E3NN. Simple E3NN with only irreducible representations of rotation order zero or 1 is not powerful enough. Using representations with rotation order of 2 will boost the performances significantly. Given the not significant performance gains beyond e3nn, further implementation details is needed.


**Questions:**

As listed in above weaknesses.

**Limitations:**

They listed the limitations in the main paper.

---

> ### Author Rebuttal · Authors · 2023-08-09
>
> Thank you for your valuable comments on our work and for recognizing that our work is the first work that integrates energy level as input! We are more than willing to address each of the specific weaknesses and questions in a detailed manner.
>
> **W1 (Limited novelty & ablation studies).**
>
> **[Regarding the novelty of the paper]**
> This paper makes two contributions: firstly, as acknowledged by the reviewer, we adopt energy embeddings as the model input instead of directly predicting densities for different energy levels; secondly, we employ prompts to capture structural-specific interactions between crystalline materials and energy levels.
>
> As we noted in lines 267-270, the integration of energy embeddings in E3NN, the best baseline method, does not yield significant benefits. This could be attributed to the potential mismatch between the intended purpose of E3NN and the use of energy embeddings. That is, the inclusion of energy embeddings may interfere with the proper learning of material “equivariance” in E3NN, leading to limited advantages in its case. However, it is important to highlight that energy embeddings continue to deliver consistent performance improvements for other models, such as MLP and Graph Networks, showcasing the effectiveness of considering energy levels as input, and applicability in a broader context.
>
> **[Regarding the ablation study of the proposed method]**
> Upon the reviewer's request, we performed ablation studies to explore the benefit of point-wise prediction of energies as in our proposed method compared with directly predicting a list of energies. To do so, we made 2 ablation models as follows (The experimental results are presented in Table 2 in the attached [PDF](https://openreview.net/forum?id=2lWh1G1W1I&noteId=YWxXBLLMtv)):
>
> - Ablation 1: We removed the Transformer architecture and utilized only energy embeddings $\mathbf{E}^{0}$, crystal representation $\mathbf{g}_i$, and crystal system prompts $\mathbf{P}$ as input for an MLP to predict the list of densities across different energy levels.
>
> - Ablation 2: While keeping the transformer architecture, we obtained crystal representation by pooling energy representations $\mathbf{E}^{L_3, i}$, and then making a direct list-wise prediction with the pooled representation.
>
> We have observed a significant drop in the model performance when excluding the transformer layers (Ablation 1), which highlights the crucial role of these layers in enabling effective point-wise interactions between atomic and energy embeddings. Furthermore, the direct list-wise prediction through pooled representation (Ablation 2) yields inferior results compared to DOSTransformer. This underscores the significance of the Transformer layer and reinforces the need to maintain its point-wise prediction mechanism for the DOS prediction task. We sincerely appreciate your insightful feedback and will make sure to incorporate these supplementary experiments into the ablation study section.
>
> **W2 (Comparison with previous crystal neural networks).**
>
> We fully agree with the reviewer that comparing DOSTransformer to recent Crystal Neural Networks can further enhance the quality of the paper. Therefore, we conducted additional experiments on Matformer, ALIGNN, and CGCNN, by adjusting the output dimensions of these models (51 for Phonon DOS and 201 for Electron DOS), in Table 3 in the attached [PDF](https://openreview.net/forum?id=2lWh1G1W1I&noteId=YWxXBLLMtv).
>
> Although both ALIGNN and Matformer exhibited commendable performance, we observe that DOSTransformer consistently surpassed them by a significant margin. These observations again demonstrate the importance of considering energy levels for accurate DOS prediction. We will make sure to include the results in the revised paper, which would undoubtedly elevate the quality of our work. We sincerely thank the reviewer for your valuable insights.
>
> **W3 (Missing implementation details of E3NN).**
>
> First of all, we apologize for any confusion caused by not elaborating on the implementation details on E3NN. We utilize the most recent code of E3NN [1] provided by the authors in the GitHub repository, and this implementation uses representation with a rotation order of 1. On the other hand, upon the reviewer’s request, we conducted experiments with the E3NN using a representation with a rotation order of 2 in Table 4 of the attached [PDF](https://openreview.net/forum?id=2lWh1G1W1I&noteId=YWxXBLLMtv).
> In the experiment, we did not observe substantial performance improvement with a rotation order of 2. The limited expressiveness of its radial functions and spherical harmonics may have contributed to this result. Considering that DOS prediction is inherently more complex than other material properties due to its sequential nature, we conjecture that this inherent complexity might have hindered potential performance improvements.
>
> [1] Chen, Zhantao, et al. "Direct prediction of phonon density of states with Euclidean neural networks." Advanced Science 8.12 (2021): 2004214.

---

> > ### Comment · Reviewer_pieM · 2023-08-10
> > **Responses to rebuttal**
> >
> > Thank you for your effort. The additional ablation studies and more comparisons with recent crystal neural networks enhanced the clarity and addressed my concerns. Therefore, I adjusted my score accordingly.
> >
> > It seems we cannot adjust the original review, but I will adjust my score to 6 if editable.

---

> > > ### Author Response · Authors · 2023-08-10
> > > **Response by authors**
> > >
> > > We thank the reviewer for acknowledging our effort, and for deciding to raise the score. We greatly appreciate it!

---

### Official Review · Reviewer_E5Yg · 2023-07-07

**Soundness:** 3 good
**Presentation:** 4 excellent
**Contribution:** 2 fair
**Rating:** 6
**Confidence:** 3

**Summary:**

This paper proposes a prompt-based Transformer network for predicting the density of states of crystalline materials. The prompts are used to represent and control the energy and the additional structure information of the materials. Experiments show the proposed method can perform very well on two datasets under different settings.

**Strengths:**

- The whole system and method are motivated well and designed carefully. The model design considers the characteristics of both the task and the deep learning methods, e.g., prompt-based Transformer.
- The experiments are carefully designed to show the benefits of the proposed methods.
- The paper is well-written and easy to read.

**Weaknesses:**

- The datasets and experiments are a bit simple. The variations of the material type and structure type in the datasets are restricted. In this sense, although prompts are used to formulate the general variation of the energy and structure, the datasets contain very simple and limited variations. It is a bit limited in evaluating the potential of the proposed method and the techniques on more realistic datasets and applications.
    - For example, the prompt is used to represent the structure. There are only seven discrete options that will be given during testing. May the model handle the structure variation without relying on the input during testing?
- In Table 4, the prompt based method is compared with the one-hot based input. Is the one-hot binary encoding directly used to replace the embedding of the prompt? If so, it might be unfair. How does the one-hot encoding work with a fully connected linear layer (i.e., feeding the one-hot encoding as the input to the FC layer), where the weights in the FC layer are corresponding to the embeddings for each type of the system?
    - In the worst case, the proposed prompt based method may work similarly to the linear embedding based approaches, on the simple datasets.

- The authors may analyze the learned prompts to indicate whether the leaned embeddings can reflect and are consistent with the physical characteristics of the system types or structures indicated by the prompts, such as whether some similarity and relationship between these characteristics can be reflected in the learned prompts.

- It seems that fine-tuning do not improves the performance significantly, especially given the identical MAE performance comparing Table 2 and Table 3.

**Questions:**

Questions are left with the weakness points.

**Limitations:**

The limitations are discussed and can reflect the main limitations of the proposed method –  the proposed method cannot encode the properties of both materials into a single model.

---

> ### Author Rebuttal · Authors · 2023-08-09
>
> **W1.**
> As the reviewer pointed out, using only 7 structural systems may make the dataset seem limited. However, the reason for having 7 structural systems is not a limitation of the dataset but rather a reflection of the knowledge used in crystallography when classifying the structures of materials [1, 2]. In other words, regardless of which dataset is used, there are only 7 crystal systems in the real-world, implying that we are bound to use the 7 crystal system prompts. Also, it is worth noting that the dataset used in the paper, i.e., the Materials Project database, is one of the most extensive public databases in the field of materials science.
>
> On the other hand, in real-world scenarios, it is widely acknowledged that databases based on DFT calculations have certain limitations in coverage. These databases often concentrate on particular material types or structural archetypes, leading to biased distributions, as discussed in lines 288-290. To address this concern, we performed experiments on out-of-distribution data in Section 5.3, where both the training and test datasets consisted of materials with different crystal systems. These experiments effectively showcase the adaptability of DOSTransformer to various dataset variations, successfully alleviating the reviewer's concerns.
>
> [1] PARK, Woon Bae, et al. Classification of crystal structure using a convolutional neural network. IUCrJ, 2017.
>
> [2] YU, Rong, et al. Calculations of single-crystal elastic constants made simple. Computer physics communications, 2010.
>
> **W2.**
> Yes, one-hot encoding is directly used to replace the embedding of prompts. As described in lines 207-210, crystal system information $P_k$ is concatenated with material embeddings $g_i$ and material-specific energy embedding $E_j^L$ and linearly transformed by the function $\phi_2$.
> Therefore, the main distinction between using Prompts and One-hot vectors lies in how the information about the crystal system is incorporated into the model, either in a soft or hard manner. When using a one-hot vector to represent the crystal system, the model integrates the crystal system information in a hard manner through matrix multiplication.
>
> In Table 4, we demonstrated that when incorporating crystal system information into “atom features”, the one-hot encoding of the system performs better than soft prompts. This is because soft prompts may hinder the learning of shared information among various crystalline materials due to their higher number of parameters. However, when incorporating the crystal system information “before the self-attention layers”, the model benefits from soft prompts. This allows the model to learn more flexible information that is advantageous for the system-specific integration of energies and crystalline materials. In conclusion, by incorporating soft prompts in appropriate locations, we were able to complete the optimal model.
>
> **W3.**
> To begin with, our objective in employing crystal prompts is not to comprehend the correlations among crystal systems, but instead to acquire crystal system-specific interactions between a crystalline material and energy levels, aiming to enrich the expert knowledge associated with each distinct crystal system like expert models.
>
> Furthermore, within the domain of crystallography, due to the unique and distinct characteristics of each crystal system in terms of symmetry, establishing a straightforward relationship between different crystal systems is not trivial [1]. However, as per the reviewer's suggestion, we conducted specific case studies by assessing the cosine similarity among the crystal system prompts based on Figure 2 in the attached [PDF](https://openreview.net/forum?id=2lWh1G1W1I&noteId=YWxXBLLMtv).
>
> Case 1) The correlation between the Cubic and Trigonal systems is approximately nine times stronger compared to the correlation between the Cubic and Triclinic systems. For example, consider the case of Bismuth (Bi). In ambient conditions, Bismuth exists in the Trigonal crystal system. However, when subjected to high-pressure conditions, it transforms into the (body-centered) Cubic crystal system. However, the Cubic and Triclinic systems exhibit the most significant dissimilarities in terms of symmetry, including axial lengths and angles, which makes crystal system transitions to be hard.
>
> Case 2) Furthermore, it is noticeable that the correlation between the triclinic and monoclinic systems is roughly ten times more pronounced than the correlation between the triclinic and cubic systems. Albite, a plagioclase feldspar mineral, exemplifies this scenario as its crystal symmetry can shift from triclinic to monoclinic contingent upon temperature changes. Nevertheless, due to the considerable discrepancy in symmetry between the triclinic and cubic systems compared to that between the triclinic and monoclinic systems, the likelihood of materials demonstrating both symmetries is quite low.
>
> To sum up, despite the complexity of establishing a direct correlation between crystal systems, our observations indicate that the alignment between prompt relationships and crystallography domain knowledge holds true.
>
> [1] Thomas, J.C., et al., A. Comparing crystal structures with symmetry and geometry. npj Comput Mater 7, 164 (2021).
>
> **W4.**
> As mentioned in lines 320-322, the performance improvement during fine-tuning is constrained due to the scarcity of data used in the process. This situation reflects real-world material sciences where existing databases relying on DFT calculations have limited coverage, often emphasizing specific material types or structural archetypes, resulting in a biased distribution. And indeed, we have conducted experiments on various training ratios for fine-tuning in Appendix E.4, revealing that as the volume of training data increases, there is a noticeable enhancement in performance during fine-tuning.

---

> > ### Comment · Reviewer_E5Yg · 2023-08-21
> > **Response to rebuttal**
> >
> > Thanks for the authors' response. Although the discussions on the very empirical observations in response to 'W2' cannot fully convince me, I feel the response addresses most of my concerns. I increased the score to 6.

---

> > > ### Author Response · Authors · 2023-08-21
> > > **Response by Authors**
> > >
> > > We sincerely appreciate the reviewer’s acknowledgement of our efforts to address the concerns and their decision to raise the score.

---

### Official Review · Reviewer_pHUW · 2023-07-11

**Soundness:** 3 good
**Presentation:** 4 excellent
**Contribution:** 4 excellent
**Rating:** 7
**Confidence:** 4

**Summary:**

The paper proposes a new transformer-based method DOSTransformer for predicting density of states of crystalline materials. Different from previous methods, the energy level is additionally modeled as an input modality, i.e. the model takes in material configuration and energy level as input to predict DOS(material, energy). DOSTransformer uses cross-attention layers to relate the material and energy representations. Self-attention layers are also used to integrate information across different energy levels. Learnable prompts are introduced to provide information about the crystal system type, which helps capture structure-specific interactions between material and energy. Experiments on phonon and electron DOS datasets show DOSTransformer outperforms existing methods like graph networks and E3NN, in both in-distribution and out-of-distribution scenarios. Ablations and sensitivity analysis demonstrate the benefits of the different components like energy embeddings, global/system losses and crystal system prompts.

**Strengths:**

- Novel problem formulation: The authors provide a new perspective on DOS prediction by treating it as a multi-modal learning problem with material and energy as inputs. This proves to work better for DOS prediction compared to prior material-only models.
- Neural net design: The multi-modal transformer architecture with cross-attention and prompt-guided self-attention is well-suited for the problem. It allows capturing complex relationships between material and energies.
- Empirical results: The model is evaluated extensively on two datasets — Phonon DOS and Electron DOS — for both in-distribution and out-of-distribution scenarios. The physical validity of predictions is also analyzed through derived material properties. In particular, the introduced energy embeddings are shown to be useful in reducing prediction errors.
- Ablation studies: The contributions of different components like the dual losses and crystal system prompts are quantitatively demonstrated through ablation studies.

**Weaknesses:**

- Simple prompts: The crystal system prompts are relatively simple vector embeddings. More sophisticated prompt learning methods could be explored to reason about structural information of different systems.

**Questions:**

N.A.

**Limitations:**

N.A.

---

> ### Author Rebuttal · Authors · 2023-08-09
>
> Thank you for your valuable comments on our work and for acknowledging the novelty of our work in DOS prediction! We are more than willing to address each of the specific weaknesses and questions in a detailed manner.
>
> **W1 (Simple prompts).**
>
> As pointed out by the reviewer, our crystal system prompts are implemented using relatively straightforward methods. We fully agree with the reviewers that we can enhance crystal system prompts in various ways. One possible direction would be integrating them with large language models (LLMs). For instance, an approach could involve initializing the system prompts using the embeddings derived from the descriptions of each crystal system.  We appreciate the valuable feedback that sheds light on potential paths for future research!

---

> > ### Comment · Reviewer_pHUW · 2023-08-18
> >
> > Thanks for the authors' response. I have also read other reviews. I agree with reviewer 9USD that it is worth adding more discussion and literature about using discretization at energy levels.

---

> > > ### Author Response · Authors · 2023-08-20
> > >
> > > We greatly appreciate your effort in reviewing our responses and consistently supporting our research paper! As we respond to reviewer 9USD, we will make sure to incorporate the discussion and literature about discretization of energy levels.

---

### Official Review · Reviewer_xyRJ · 2023-07-12

**Soundness:** 3 good
**Presentation:** 3 good
**Contribution:** 3 good
**Rating:** 6
**Confidence:** 4

**Summary:**

This work proposes a transformer architecture for predicting density of states of crystalline materials for different energy levels.
The distribution of states is spectral property that is approximated as a function of both the material and energy levels.
The architectures proposed consists of a multi-modal transformer with self and cross attention layers, and a GNN for embedding.
Experiments compare the proposed architecture with state-of-the-art E3NNs, demonstrating a decrease in MSE.

**Strengths:**

The main strengths of this work include:
1. The approach considers the crystalline material, energy levels, and structural properties.
2. The problem and methods are well-defined, and the architectures are implemented using state-of-the-art libraries.
3. The results include out-of-distribution experiments, ablation studies, and sensitivity analysis.

**Weaknesses:**

The main weaknesses of this work are that:
1. It needs to be clarified what MSE's are required for the approach to be useful in the real world.
For example, in proteomics, and RMSD of less than 2-3Angstrom is considered valid compared to experimental results.
What would be the MSE's required to achieve such a validation in this domain.
2. The terminology and definition used for fine-tuning and especially prompting in this work differs significantly from the standard definition
and should be clarified.

**Questions:**

What are the MSE values of bulk, band, and ferm. are considered useful for real-world applications?
Are there 38,889 crystalline materials after or before filtering for non-magnetism?

**Limitations:**

The limitations are adequately addressed.

---

> ### Author Rebuttal · Authors · 2023-08-09
>
> Thank you for your valuable comments on our work and for acknowledging the efforts in experiments, including out-of-distribution scenarios! We are more than willing to address each of the specific weaknesses and questions in a detailed manner.
>
> **W1, Q1 (Appropriate level of MSE).**
>
> To the best of our knowledge, in the literature of DOS prediction, deriving various physical properties based on the model-predicted DOS is a novel endeavor, making it challenging to determine a definitive threshold for the MSE value that is practically applicable in the real world.
>
> For band gap prediction, a commonly accepted MSE value is around 0.3 ~ 0.4 [1]. However, we would like to emphasize that a direct comparison of our results with those presented in [1] is not appropriate considering our experimental setup.
>
> - **Experimental setup:** As described in Line 277-282, we used the DFT-calculated DOS as the ground-truth DOS, and trained a simple 2-layer MLP with non-linearity to predict the properties of a crystal structure. Then, we use the learned MLP weights to simply predict various physical properties given the model-predicted DOS as input.
>
> It is important to note that the learned MLP weights themselves are not perfect, meaning that the properties derived from the model-predicted DOS using the learned MLP weights inevitably exhibit error that is accumulated due to the error in the learned MLP weights. For this reason, rather than directly comparing our results with the results reported in [1], our intention was to compare among our baselines, and see whether the various properties predicted by DOS predicted by our proposed method are relatively more accurate. Thus, in the current context, the MSE value itself might hold less significance, and it is more appropriate to consider it as an evaluation metric to compare among baselines. However, we could reduce the accumulated error by training a more advanced neural network instead of a simple MLP. We appreciate the valuable feedback that sheds light on potential paths for future research!
>
> [1] LEE, Joohwi, et al. Prediction model of band gap for inorganic compounds by combination of density functional theory calculations and machine learning techniques. Physical Review B, 2016, 93.11: 115104.
>
> **W2 (Terminology of fine-tuning and prompt-tuning).**
>
> Although we made efforts to clarify the concepts of fine-tuning and prompt tuning in Section 2.2, we apologize for any confusion that may have arisen. To provide further clarification, we briefly outline the distinctions between fine-tuning, discrete prompt designing, and continuous prompt tuning.
>
> In the field of Natural Language Processing (NLP), fine-tuning was introduced to adapt pre-trained language models (LM) for specific downstream tasks. Conversely, discrete prompt designing aims to reformulate downstream tasks to resemble those encountered during the original LM training, achieved by using a textual prompt. Recently, continuous prompt tuning has been proposed, which involves prompting directly in the embedding space of the model. These prompts have their own parameters, which can be adjusted based on training data from the downstream task. [1]
>
> In this paper, we adopt the concept of continuous prompt tuning to guide the model in understanding the structural types of crystalline materials. While a single prompt was employed for each downstream task in NLP, our approach utilizes a single prompt for each of the seven structural types of materials discussed in the paper. It is important to note that continuous prompts are utilized due to the non-trivial nature of modeling the structural type of crystalline material using human-engineered prompts.
>
> Additionally, concerning the use of "fine-tuning" in Section 5.3 and Table 3, we intended it to refer to tuning on various downstream tasks. In Table 3, "All" denotes the model that updates all parameters for the downstream task, whereas "Only Prompt" refers to the model that exclusively updates the parameters of continuous prompts, which is the same as "Continuous prompt tuning." We will thoroughly revise the manuscript to ensure clarity on these matters.
>
> [1] LIU, Pengfei, et al. Pre-train, prompt, and predict: A systematic survey of prompting methods in natural language processing. ACM Computing Surveys, 2023, 55.9: 1-35.
>
>
> **Q2 (Non-magnetism?).**
>
> Yes, 38,889 crystalline materials are non-magnetic materials obtained after filtering out magnetic materials. This choice was influenced by the well-known issue of calculation accuracy in the DFT-generated DOS of magnetic materials, which is considered unreliable [1, 2]. In short, we only used non-magnetic materials for training our model, since using magnetism materials for training may interfere with the model's ability to learn the DOS of non-magnetic materials.
>
> [1] Zeller, R. (2006). Spin-polarized dft calculations and magnetism. Computational Nanoscience: Do It Yourself, 31, 419-445.
> [2] Poblet, J. M., López, X., & Bo, C. (2003). Ab initio and DFT modelling of complex materials: towards the understanding of electronic and magnetic properties of polyoxometalates. Chemical Society Reviews, 32(5), 297-308.

---

### Official Review · Reviewer_9USD · 2023-07-19

**Soundness:** 3 good
**Presentation:** 3 good
**Contribution:** 3 good
**Rating:** 7
**Confidence:** 2

**Summary:**

This work proposes a transformer model to predict the density of states of given materials. The model takes in all energy levels that are desired for the Density of States calculation as a 'prompt' and information about the materials structure, the output is a prediction of the DOS at the given energy levels.

**Strengths:**

+ The DOSTransformer is evaluated on both in-domain structures and out of domain structures against relevant baseline models and previously reported models.
+ The DOSTransformer model design makes use of domain specific knowledge, such as the crystal structure type and the energy levels.
+ Authors demonstrate that the task of predicting DOS is heavily added by providing energy levels for targeted prediction via relevant ablation studies
+ The model is clearly described. I was able to understand all the components of the proposed model.

**Weaknesses:**

- More could be done to describe the task of predicting DOS for downstream applications, especially for a non-materials science audience. How are the DOS outputs for predicting bandgap energy / electrical conductivity? How many grid points are needed to gain an 'accurate enough' prediction of these properties? What is kind of error in the DOS prediction is reasonable? Are there certain energy levels where it is more critical to predict the DOS correctly?

- I would appreciate some more statistics of the datasets used (PhononDOS and ElectronDOS) in the main text (e.g. # atom types, # crystal types).

**Questions:**

- As I understand it one of the tasks in this work is to predict the output of a function (Density of States) on a fixed grid of energy levels. Are there other examples in the literature where transformers are used to predict the outputs of a mathematical function, and the fixed grid of inputs is provided as a prompt for the model? If there are, it would be great to learn about them in the background section.

- Because the DOS is predicted over a finite grid, I am curious about the smoothness of the DOSTransformer model between finite grid points. If the DOSTransformer model was to be evaluated at energy levels between the provided finite points in the prompt, are the predictions smooth where you'd expect the function to be smooth?

**Limitations:**

- More description should be provided about the datasets used. How many different elements/ types of materials were included int he PhononDOS and ElectronDOS provided?

---

> ### Author Rebuttal · Authors · 2023-08-09
>
> Thank you for your valuable comments on our work. We are more than willing to address each of the specific weaknesses and questions in a detailed manner.
>
> **W1 (Downstream tasks of DOS prediction).**
> - How are the DOS outputs for predicting bandgap energy / electrical conductivity?
>
> DOS serves as a fundamental representation of the electronic density in an atomic system across various energy states. As the electronic density directly influences the electric charge and physical energy of the atoms, DOS provides valuable insights into material properties, including band gap and electrical conductivity. For instance, band gap, a crucial factor in determining the material's chemical applications, can be derived by examining the highest and lowest electronic densities in the DOS.
>
> - How many grid points are needed to gain an 'accurate enough' prediction of these properties?
>
> The electron distributions vary for each material, and the number of required grid points depends on these differences. Due to the variations, we adopted the DFT configurations from the Materials Project database, one of the most extensive public databases in the field of materials science.
>
> - What kind of error in the DOS prediction is reasonable?
>
> We mainly compare DOSTransformer to baselines in terms of MSE and MAE as have done in previous works [1]. In addition to the metrics, we further introduced R2, which is crucial in evaluating the regression models, by indicating the proportion of the variance in the dependent variable (i.e., DOS) that is predictable from independent variables (i.e., crystalline materials and energies).
>
> Moreover, in terms of physics, the density peaks within the DOS carry significance beyond the entirety of the density sequence across energy states. In other words, it's important to predict the energy level at which the electron density is the highest. To address this, we introduce the absolute error of peak positions as an extra measure to evaluate the model's prediction accuracy. You can find this in Table 1 in the attached [PDF](https://openreview.net/forum?id=2lWh1G1W1I&noteId=YWxXBLLMtv) .
>
> In the table, we observe that DOSTransformer provides more precise predictions for the peak points in the DOS, showcasing the practical value of the predicted DOS.
>
> [1] CHEN, Zhantao, et al. Direct prediction of phonon density of states with Euclidean neural networks. Advanced Science, 2021, 8.12: 2004214.
>
> - Are there certain energy levels where it is more critical to predict the DOS correctly?
>
> Important energy levels are different for each material. However, the energy levels of the highest electronic densities are important because they determine many energy-related properties of the materials. For this reason, we evaluated how accurately the predicted peak points in the DOS match the actual peak points in the real DOS by comparing them in terms of MSE in Table 1 in the attached [PDF](https://openreview.net/forum?id=2lWh1G1W1I&noteId=YWxXBLLMtv) .
>
>
> **W2 (Dataset statistics).**
>
> In addition to Electron DOS data statistics provided in Appendix A.3., we provide more details in the attached [PDF](https://openreview.net/forum?id=2lWh1G1W1I&noteId=YWxXBLLMtv) .
>
>
> **Q1 (Literature survey).**
>
> Firstly, we would like to provide clear definitions of the terminology used in this paper. We refer to the embeddings for each fixed grid of energy as "energy embeddings," and the prompts for each crystal system as "crystal system prompts." When conceptualizing each energy value as the position of a word in a sentence, energy embeddings can be analogous to the positional encoding used in traditional transformers.
>
> In the original proposal of the Transformer architecture, a sinusoidal function was suggested as a method for positional encoding. However, the sinusoidal function may have limitations in terms of learnability and flexibility, which can affect its effectiveness [1].
>
> To address this issue, most pre-trained language models [2, 3] employ learnable vector embeddings as positional representations. Additionally, [1] enhances positional encoding by constructing a learnable fully-connected feed-forward sinusoidal positional encoding network.
>
> We apologize for any confusion that may have arisen as a result of the missing background section. To address this, we will make comprehensive revisions to the section by incorporating the background about positional features.
>
> [1] Guoxin Wang, et al. A Simple yet Effective Learnable Positional Encoding Method for Improving Document Transformer Model. ACL Findings 2022.
> [2] DEVLIN, Jacob, et al. Bert: Pre-training of deep bidirectional transformers for language understanding. ACL 2019.
> [3] LIU, Yinhan, et al. Roberta: A robustly optimized bert pretraining approach. arXiv preprint arXiv:1907.11692, 2019.
>
>
> **Q2 (Smoothness of DOS prediction).**
>
> As pointed out by the reviewer, we transformed the continuous energy values into finite grids of energy levels through binning. This discretization process prevents the evaluation of DOS prediction models, including DOSTransformer, at energy levels between the provided finite points. However, for specific ranges of energies with desired resolutions, we have the flexibility to train the model with the newly processed data. As a result, we believe that DOSTransformer can be applied universally to accommodate different energy levels and resolutions according to specific research needs.
>
> On the other hand, a promising avenue for DOS prediction, considering the continuous nature of energies and DOS, could be the incorporation of neural ordinary differential equations (Neural ODEs) [1]. Neural ODEs have shown effectiveness in capturing the continuous nature of sequential data, making them a potential candidate for this task. We are grateful for the insightful feedback, which illuminates future research directions!
>
> [1] Chen, Ricky TQ, et al. "Neural ordinary differential equations." NeurIPS 2018

---

> > ### Comment · Reviewer_9USD · 2023-08-15
> >
> > I have read the reviewer's responses. Thank you to the reviewers for their careful responses and improvements to the manuscript!

---

### Author Rebuttal · Authors · 2023-08-09

Dear reviewers, thank you for your valuable comments on our work. We are more than willing to address each of the weaknesses and questions in detail. We also attach a PDF file that contains Figures and Tables for rebuttal.

---

### Decision · Program_Chairs · 2023-09-21

**Decision:**

Accept (poster)

**Comment:**

The paper proposes a multi-model transformer to estimate the density of states of crystalline materials.

The paper received five reviews and all reviewers are in favor of accepting the paper.

The reviewers acknowledge that the problem is novel and interesting, the design of the network is well-suited for the problem, and that the method is evaluated well (both in- and out-of-domain), and that it includes ablation studies and sensitivity analysis. Moreover, the paper is well written.

The reviewers also find a few minor weakness that could for the most part clear up in the rebuttal and/or can be incorporated in the final version.